# Near-Infrared Imaging of Colonic Adenomas In Vivo Using Orthotopic Human Organoids for Early Cancer Detection

**DOI:** 10.3390/cancers15194795

**Published:** 2023-09-29

**Authors:** Xiaoli Wu, Chun-Wei Chen, Sangeeta Jaiswal, Tse-Shao Chang, Ruoliu Zhang, Michael K. Dame, Yuting Duan, Hui Jiang, Jason R. Spence, Sen-Yung Hsieh, Thomas D. Wang

**Affiliations:** 1Division of Gastroenterology, Department of Internal Medicine, University of Michigan, Ann Arbor, MI 48109, USA; wxiaoli@med.umich.edu (X.W.); jaiswals@med.umich.edu (S.J.); mdame@med.umich.edu (M.K.D.); spencejr@med.umich.edu (J.R.S.); 2Department of Gastroenterology and Hepatology, Linkou Chang Gung Memorial Hospital, Taoyuan 33305, Taiwan; 8902088@cgmh.org.tw; 3Department of Mechanical Engineering, University of Michigan, Ann Arbor, MI 48109, USA; tsechang@umich.edu; 4Department of Biomedical Engineering, University of Michigan, Ann Arbor, MI 48109, USA; rlzhang@umich.edu; 5Department of Biostatistics, University of Michigan, Ann Arbor, MI 48109, USA; yutingd@umich.edu (Y.D.); jianghui@umich.edu (H.J.)

**Keywords:** imaging, cancer, early detection, cMet, peptide, fluorescence, colonoid

## Abstract

**Simple Summary:**

Colorectal cancer is a leading cause of worldwide cancer-related morbidity and mortality. Conventional white light colonoscopy is routinely used to identify and remove colonic polyps. However, premalignant lesions that are flat and subtle in appearance are often missed. Cells alter in molecular expression well in advance of morphological changes. cMet is highly overexpressed by colonic adenomas. We aim to implant patient-derived organoids in the colon of immunocompromised mice to demonstrate a model system for evaluating the specific uptake of a near-infrared labeled peptide ligand. Human adenoma and normal colonoids were imaged in vivo using a small animal endoscope. A high target-to-background ratio was observed ~1 h post-injection followed by clearance after ~24 h. The mean intensity measured was significantly greater for the adenoma versus normal colonoids. These results demonstrate promise for a targeted imaging approach using fluorescently labeled peptides to identify precancerous lesions that would otherwise be missed using white light illumination alone.

**Abstract:**

Colorectal cancer is a leading cause of cancer-related morbidity and mortality worldwide. Premalignant lesions that are flat and subtle in morphology are often missed in conventional colonoscopies. Patient-derived adenoma colonoids with high and low cMet expression and normal colonoids were implanted orthotopically in the colon of immunocompromised mice to serve as a preclinical model system. A peptide specific for cMet was labeled with IRDye800, a near-infrared (NIR) fluorophore. This peptide was administered intravenously, and in vivo imaging was performed using a small animal fluorescence endoscope. Quantified intensities showed a peak target-to-background ratio at ~1 h after intravenous peptide injection, and the signal cleared by ~24 h. The peptide was stable in serum with a half-life of 3.6 h. Co-staining of adenoma and normal colonoids showed a high correlation between peptide and anti-cMet antibody. A human-specific cytokeratin stain verified the presence of human tissues implanted among surrounding normal mouse colonic mucosa. Peptide biodistribution was consistent with rapid renal clearance. No signs of acute toxicity were found on either animal necropsy or serum hematology and chemistries. Human colonoids provide a clinically relevant preclinical model to evaluate the specific uptake of a NIR peptide to detect premalignant colonic lesions in vivo.

## 1. Introduction

Colorectal cancer (CRC) is the third most common cancer in incidence and the second-leading cause of cancer-related deaths worldwide [1,2,3]. New cases of CRC are increasing at a rapid rate in transitioning countries and are being seen more often in the young adult population [4]. Colonoscopy has been recommended by major medical societies as the primary method to screen for CRC in patients at increased risk. This minimally invasive in vivo imaging procedure has been widely accepted by the general patient population and is the preferred approach for CRC screening [5]. However, improved techniques are needed for early cancer detection, in particular, in the proximal (right) colon [6]. Conventional colonoscopy uses white light illumination to identify the presence of structural lesions within colonic mucosa, such as masses and polyps. This modality relies on slight differences in reflected light and has intrinsic limitations in image contrast. As a result, >25% of adenomas [7,8,9] and an unknown percentage of flat lesions go unidentified [10]. Also, subtle lesions and diminutive polyps, which have an increased risk for cancer transformation, can be easily missed. Interval cancers occur when CRC arises within 5 years after a completed exam, and they are increasing in incidence [11,12,13]. Thus, improved methodologies for early cancer detection during a colonoscopy are needed to reduce the global burden of this disease.

Molecular targets are expressed by cells well before structural changes occur that lead to the formation of mass lesions that become grossly visible. This process provides a window of opportunity for precancerous lesions to be identified prior to the formation of frank tumors. Highly specific molecular probes, including antibodies, enzymes, small molecules, lectins, and peptides, are being developed to provide a biological basis for early cancer detection [14,15,16,17,18]. These agents are fluorescently labeled to generate images with high contrast for real-time in vivo visualization of overexpressed targets. This targeted approach provides a “red flag” for physicians to guide tissue biopsy and resection. The mesenchymal-epithelial transition factor (cMet) consists of a 50 kDa extracellular α-subunit and a 145 kDa transmembrane β-subunit that results from the expression of the MET proto-oncogene [19]. cMet is a tyrosine kinase that is activated by hepatocyte growth factor (HGF), which stimulates the cMet/HGF signaling pathway [20]. *MET* amplification and HGF/cMet overexpression have been reported to play a key role in CRC development [21,22,23]. This aberrant signaling pathway has been associated with cell motility, proliferation, apoptosis, and angiogenesis [24,25]. A number of meta-analyses have concluded that high cMet expression was associated with significantly worse overall survival in CRC patients [26,27].

Peptides are a promising probe platform for clinical use as ligands that bind specifically to molecular targets overexpressed by premalignant lesions. These protein fragments are intrinsically high in diversity and can be designed to bind a broad range of cell surface targets with high specificity and affinity on the nanomolar scale. Peptides can be delivered effectively to the mucosal surface in the digestive tract at high concentrations to maximize target interactions and achieve rapid binding with minimal risk for toxicity. Peptides can be administered systemically for more comprehensive tissue biodistribution if serum stability is adequate [28]. Peptides have the flexibility to be labeled with a broad range of fluorophores [29] and are inexpensive to mass manufacture [30]. Peptide ligands are short in length and have low potential for immunogenicity [31,32], which allows for repetitive in vivo use. These peptide features are well suited for clinical use in procedures performed in high volumes, such as colonoscopies. Here, we aim to demonstrate the use of a peptide specific for cMet with intravenous administration to identify premalignant lesions in vivo using endoscopic imaging. Patient-derived adenoma colonoids with high and low cMet expression and a normal colonoid will be implanted orthotopically in the colon of immunocompromised mice to serve as a preclinical model for CRC.

## 2. Materials and Methods

### 2.1. Colonoid Specimens and Culture Conditions

Patient-derived colonoids were derived from specimens of human adenoma and normal colon [33,34,35]. An adenoma colonoid with high cMet expression (cMet^high^) was derived from a large adenoma associated with adenocarcinoma tissues (46 years old, male). A normal colonoid was derived from deceased donor tissue (43 years old, female). These colonoids were provided by the Translational Tissue Modeling Laboratory (TTML; University of Michigan IRB REP00000105, unregulated designation). The low cMet adenoma colonoid (cMet^low^) was derived from a pinch biopsy (64 years old, male) and was generated in our laboratory (University of Michigan IRB HUM00102771).

Cultures were grown in 6-well tissue culture plates (USA Scientific CytoOne, Ocala, FL, USA; #CC7682-7506) using Matrigel with growth media diluted to 8 mg/mL (Corning Thermo Fisher Scientific, Glendale, AZ, USA; #354234). Cultures were passaged by triturating and dissociating Matrigel in cold DPBS, centrifuging at 300× *g*, and plating with 2.5 µM CHIR99021 (Tocris Bio-Techne, Minneapolis, MN, USA; #4423) and 10 µM Y27632 (Tocris Bio-Techne, #125410). The normal colonoid was cultured in LWRN complete medium containing 50% L-WRN conditioned medium (source of Wnt3a, R-spondin-3 and Noggin), advanced DMEM/F-12 (Gibco Thermo Fisher, Grand Island, NY, USA; #12634028), N-2 media supplement (Gibco Thermo Fisher Scientific; #17502048), B-27 supplement minus vitamin A (Gibco Thermo Fisher Scientific; #12587010), 1 mM N-Acetyl-L-cysteine (Sigma-Aldrich, Burlington, MA, USA; #A9165), 2 mM GlutaMax (Gibco Thermo Fisher Scientific, #35050-061), 10 mM HEPES (Gibco Thermo Fisher Scientific; #15630080), 50 units/mL penicillin, 0.05 mg/mL streptomycin (Gibco Thermo Fisher Scientific; #15070063), 100 µg/mL Primocin (InvivoGen, San Diego, CA, USA; ##ant-pm-1), 100 ng EGF/mL (R&D Systems, Inc Bio-Techne, Minneapolis, MN, USA; #236-EG), 10 µM SB202190 (Tocris Bio-Techne; #126410), 500 nM A83-01 (Tocris-Bio-Techne, #293910), and 10 µM Y27632. The cMet^high^ adenoma colonoid was cultured in a manner similar to that for the normal colonoid but without Y-27632 and SB202190. The cMet^low^ adenoma colonoid was cultured in a manner similar to that for the normal colonoid but without SB202190.

### 2.2. Patient-Derived Colonoid Implantation/Orthotopic Xenograft Mouse Model

All animal experimental procedures were performed in compliance with relevant regulations of the University of Michigan. Mice were housed per the guidelines of the Unit for Laboratory Animal Medicine (ULAM), and in vivo imaging was performed with approval by the Institutional Animal Care and Use Committee (IACUC).

Colonoids were harvested and treated, as described previously [24,25,26,27,28,29,30,31,32]. Colonoids were resuspended in engraftment medium: DPBS, 5% Matrigel, 10 µM Y27632. Prior to colonoid implantation, 7–12-week-old NSG (NOD.Cg-*Prkdc^scid^Il2rg^tm1Wjl^*/SzJ, Jackson laboratory, #005557) mice were treated with 2.5% dextran sulfate sodium (DSS) for 5 days and supplied with regular water for another day. Anesthesia was induced and maintained via a nose cone with inhaled isoflurane mixed with oxygen at concentrations of 2–4% at a flow rate of 0.5 L/min. A total of ~2.5–5  × 10^5^ cell aggregates in 200 μL were injected intrarectally. The rectum was closed immediately using tissue adhesive (Santa Cruz Biotechnology, Dallas, TX, USA; #SC361931) to promote retention of the implanted colonoids. Colonoid growth was monitored weekly by small animal endoscopy with white light illumination. Imaging was performed after the implanted adenomas were clearly visible.

*CPC;Apc* mice have been genetically engineered to sporadically delete an *APC* gene regulated by a Cre promoter (CDX2P-9.5NLS-Cre) to spontaneously form either flat or polypoid adenomas in the distal colon [36].

### 2.3. NIR Peptide Specific for cMet

The cMet and control peptides were labeled by mixing 2 mg of peptide and 1 mg of IRDye800CW maleimide (LI-COR Biosciences, Lincoln, NE, USA) in 2 mL of coupling buffer (0.1 M sodium phosphate, 0.5 mM TCEP, pH 7.4). The reaction was performed in N_2_ for 2 h at room temperature (RT). The NIR-labeled peptides were purified using reversed-phase high-performance liquid chromatography (RP-HPLC). The crude peptides were characterized using mass spectrometry and were evaluated by HPLC for purity >99%. The peptides were lyophilized for storage at −80 °C.

The peptide absorbance spectra were measured at 10 μM concentration using a UV-Vis spectrophotometer (Thermo Scientific, #NanoDrop 2000). The peptide fluorescence emission was measured using a fiber-coupled spectrophotometer (Ocean Optics, Orlando, FL, USA, #USB2000+) with λ_ex_ = 785 nm. MATLAB R2022a (Mathworks, Natick, MA) software was used to plot the spectra.

### 2.4. Colonoid Characterization

Cultured colonoids were embedded in histogel and prepared as formalin-fixed, paraffin-embedded (FFPE) blocks. FFPE sections of colonoids and mouse colon were deparaffinized, and antigen retrieval was performed as previously described [37]. Sections were incubated overnight with 1:100 dilution of monoclonal rabbit anti-cMet antibody (Abcam, Boston, MA, USA; #EP1454Y) at 4 °C. Antigen binding was detected with a VECTASTAIN Elite ABC HRP Kit (Vectorlabs, Newark, CA, USA; #PK6101) and chromogen 3, 3′-diaminobenzidine (DAB; Sigma, #D3939) per manufacturer instructions. Slides were counterstained with Harris hematoxylin. Controls were prepared similarly without primary anti-cMet antibody (Abcam, #EP1454Y). Adjacent sections were processed for routine histology (H&E).

### 2.5. Validation of Specific Peptide Binding to Colonoids

Colonoid sections were deparaffinized, and antigen retrieval was performed as described previously [37]. Tissue sections were blocked with goat serum for 30 min at RT. Overnight incubation at 4 °C was performed with the addition of monoclonal anti-cMet antibody (Abcam, #EP1454Y) at 1:100 dilution. AF488-labeled goat anti-rabbit secondary antibody (Life Technologies, Carlsbad, CA, USA; #A-11029) was applied at 1:500 dilution. The slides were stained with peptides diluted in PBS at a concentration of 5 μM for 10 min at RT after being washed with PBST. Sections were then washed with PBST and mounted with Prolong Gold reagent containing DAPI (Invitrogen, Waltham, MA, USA; #P36931). NIR fluorescence images were collected using an inverted confocal microscope (Leica, Deerfield, IL, USA; Stellaris 5). Intensities were quantified using custom MATLAB R2022a software.

### 2.6. In Vivo Image Validation of Specific Peptide Binding

Mice were fasted for 4–6 h prior to in vivo imaging. Anesthesia was induced and maintained using inhaled isoflurane mixed with oxygen at concentrations of 2–4% and a flow rate of 0.5 L/min via a nose cone. The peptides were injected intravenously using a concentration of 200 μM diluted in 200 μL of PBS. A rigid small animal endoscope (Karl Storz Veterinary Endoscopy, Goleta, CA, USA) was modified to deliver NIR excitation at λ_ex_ = 785 nm [28]. This instrument was inserted into the mouse rectum at 1 h post-injection to collect in vivo fluorescence images. After 3 days to allow for target peptide clearance, the scrambled (control) peptide was administered systemically and imaged similarly. A total of *n* = 3 independent regions of interest (ROI) with dimensions of 20 × 20 μm^2^ were identified randomly from the implanted area (target) and from adjacent normal colonic mucosa (background). The mean fluorescence intensities were used to calculate target-to-background (T/B) ratios.

### 2.7. Pharmacokinetics

The IRDye800-labeled target peptide (200 μM, 200 μL) was injected intravenously in *Cpc;Apc* mice, and fluorescence images were collected at 0, 0.5, 1, 2, 4, 6, 10, 24, 48 h post-injection using the small animal endoscope. The IRDye800-labeled scrambled (control) peptide was administered and imaged the same way 3 days after. The adjacent nontumor tissue region with an equal area of the tumor region was used for background. The time point with the highest target-to-background (T/B) ratio was selected as peak uptake.

### 2.8. In Vivo Serum Stability

Peptide stability was evaluated by incubating 6 μL of 1 mM peptide in PBS at 37 °C, resulting in a final concentration of 30 μM in 194 μL of serum obtained from a *CPC;Apc* mouse. Samples were precipitated by adding a double volume of acetonitrile and centrifuging for 10 min at 4 °C. RP-HPLC was performed at 0, 0.5, 1, 2, 4, 8, and 24 h. The relative peptide concentrations were determined by measuring the area under the peak (Waters, Breeze 2). The coefficient of determination R^2^ was determined using Graphpad Prism 8 software (GraphPad Software, Boston, MA, USA).

### 2.9. Ex Vivo Image Validation of Specific Peptide Binding

Tissue sections were deparaffinized, and antigen retrieval was performed as described previously [37]. Sections were blocked with goat serum for 30 min at RT followed by overnight incubation with 1:100 dilution of monoclonal anti-cMet antibody (Abcam; #EP1454Y) and undiluted human-specific anti-cytokeratin (CAM 5.2, BD Biosciences, Franklin Lakes, NJ, USA; #345779) at 4 °C. Thereafter, 1:500 AF488-labeled goat anti-rabbit secondary antibody (Life Technologies; #A-11029) and 1:500 AF568-labeled goat anti-mouse secondary antibody (Abcam, #ab175473) were administered. After washing with PBST, the slides were stained with the IRDye800-labeled peptides at a concentration of 5 μM for 10 min at RT. The tissue sections were then washed with PBST and mounted using a Prolong Gold reagent containing DAPI (Invitrogen; #P36931). NIR fluorescence images were collected using an inverted confocal microscope (Leica; Stellaris 5). The fluorescence intensities were quantified using custom MATLAB R2022a, and regions of saturated intensities were avoided.

### 2.10. Peptide Biodistribution

Peptides (200 μM, 200 μL) were injected intravenously in NSG mice ~3 weeks after orthotopic implantation of the patient-derived colonoids. The mice were euthanized at 1 h post-injection. Major organs were resected and imaged using the Pearl Trilogy (LI-COR Biosciences) with excitation at λ_ex_ = 785 nm and emission at λ_em_ = 820 nm to measure the peptide biodistribution. Fluorescence intensities were quantified from each whole tumor or organ using Image Studio 5.5.4 software (LI-COR Biosciences).

### 2.11. Peptide Safety

*CPC;Apc* mice were euthanized at 48 hours post-injection with QQT* -IRDye800 (200 μM, 200 μL). Vital organs were excised, sectioned, and evaluated by routine histology (H&E) to evaluate for signs of acute toxicity. Whole blood was collected retro-orbitally using a capillary tube and submitted for evaluation of hematology and chemistry.

## 3. Results

### 3.1. NIR Peptide Specific for cMet

A 7-mer peptide with sequence QQTNWSL was identified using phage display methods [38]. The peptide (black) was synthesized and labeled with IRDye800 (red) via a GGGSC linker (blue) on the C-terminus to prevent steric hindrance (hereafter QQT*-IRDye800), as shown in Figure 1a. IRDye800 was chosen for use as the fluorophore because of its high quantum yield and photostability. The sequence was scrambled as TLQWNQS for use as control and was also labeled with IRDye800 (hereafter TLQ*-IRDye800), as shown in Figure 1b. The IRDye800-labeled peptides were purified to >95% by HPLC. An experimental mass-to-charge (*m*/*z*) ratio of 2361.80 was measured for QQT*-IRDye800 and TLQ*-IRDye800 using mass spectrometry. These results agree with expected values, as shown in Appendix A. Peak peptide absorbance and emission occur in the NIR window from ~650–900 nm where autofluorescence background and tissue scattering are substantially reduced, as shown in Figure 1c.

### 3.2. Colonoid Characterization

Patient-derived adenoma colonoids with high and low cMet expression and a normal colonoid (control) were selected. Brightfield (BF) images of each colonoid are shown in Figure 2a. Histology (H&E) from colonoid sections is shown in Figure 2b. An IHC stain confirms the expected cMet expression levels for the adenoma and normal colonoids, as shown in Figure 2c.

### 3.3. Validation of Specific Peptide Binding to Colonoids

Co-staining of adenoma and normal colonoid sections with QQT*-IRDye800 peptide and anti-cMet-AF488 antibody was performed ex vivo. Immunofluorescence (IF) was used to evaluate the strength of peptide staining to distinguish high from low cMet expression. Strong fluorescence intensity was observed from QQT*-IRDye800 and anti-cMet-AF488 for the cMet^high^ adenoma colonoid, and the reduced signal was detected with TLQ*-IRDye800, as shown in Figure 3a. Reduced fluorescence signal was seen with QQT*-IRDye800, anti-cMet-AF488, and TLQ*-IRDye800 for the cMet^low^ adenoma colonoid, as shown in Figure 3b, and for the cMet^low^ normal colonoid, as shown in Figure 3c. The merged image shows the co-localization of peptide and antibody binding with high correlation in all specimens. The mean (±SD) values were significantly greater for QQT*-IRDye800 and anti-cMet-AF488 for the cMet^high^ adenoma colonoid versus the cMet^low^ adenoma and normal colonoids, as shown in Figure 3d. The differences with TLQ*-IRDye800 for all colonoids were nonsignificant.

### 3.4. In Vivo Image Validation of Specific Peptide Binding

Patient-derived colonoids were implanted orthotopically in the colon of immunocompromised mice. A rigid small animal endoscope adapted to collect NIR fluorescence was used to collect in vivo images at ~3 weeks following implantation. Viable implanted lesions (arrows) were seen in the mouse colon using white light (WL) illumination. Strong fluorescence intensity was observed for the cMet^high^ adenoma colonoid at 1 h after intravenous administration of QQT*-IRDye800, as seen in Figure 4a. The same lesion showed minimal signal with TLQ*-IRDye800. Reduced fluorescence intensity was observed from the cMet^low^ adenoma colonoid for both peptides, as seen in Figure 4b. Minimal fluorescence intensity was observed from the normal colonoid for both peptides, as shown in Figure 4c. The fluorescence intensities were quantified, and the mean (±SD) value for QQT*-IRDye800 was significantly greater than that for TLQ*-IRDye800 with the cMet^high^ adenoma colonoid, as seen in Figure 4d. The differences in mean values for the cMet^low^ adenoma and normal colonoids were nonsignificant.

### 3.5. Pharmacokinetics

NIR fluorescence images were collected to evaluate the time course for in vivo peptide uptake by colonic adenomas, as shown in Appendix A. Minimal intensity was seen from the tumors prior to peptide injection (0 h). QQT*-IRDye800 and TLQ*-IRDye800 were administered intravenously, and images were collected over 48 h (Figure 5a). Quantified intensities showed a peak T/B ratio at 1 h after injection of QQT*-IRDye800. The signal decreased to the baseline by ~24 h post-injection.

### 3.6. In Vivo Serum Stability

The stability of QQT*-IRDye800 in mouse serum over 24 h was evaluated using analytical RP-HPLC. The relative concentration was determined by the area under the peak (Appendix A), and a half-life of T_1/2_ = 3.6 h was measured (Figure 5b).

### 3.7. Ex Vivo Image Validation of Specific Peptide Binding

Mice were euthanized after completion of imaging, and the colon was excised and sectioned. Ex vivo immunofluorescence staining was performed to further validate specific peptide binding. QQT*-IRDye800 (red) and anti-cMet-488 (green) showed strong fluorescence signals in human (h) regions occupied by the cMet^high^ adenoma by comparison with that for adjacent normal mouse (m) colonic mucosa, as shown in Figure 6a. A human-specific cytokeratin (h-CKT, cyan) stain was used to identify the boundary (dotted line) between the two tissue types. Reduced fluorescence intensity was seen with QQT*-IRDye800 (red) and anti-cMet-488 (green) from the implanted cMet^low^ adenoma, as shown in Figure 6b. No difference in signal was seen from the adjacent normal mouse colon. Similar results were observed for the normal colonoid (Figure 6c). Quantified fluorescence intensities showed a significantly greater mean (±SD) value for QQT*-IRDye800 and anti-cMet-AF488 for the cMet^high^ adenoma versus the cMet^low^ adenoma and normal colonoid (Figure 6d). No significant difference in mean intensities was found for antihuman cytokeratin.

### 3.8. Peptide Biodistribution

After completion of imaging, the animals were euthanized, and white light and NIR fluorescence images were collected from major organs and were shown as an overlay in Appendix A. Specific uptake of QQT* -IRDye800 by the cMet^high^ adenoma colonoid (arrow) was found to be significantly higher than that for cMet^low^ (Appendix A). Reduced signal was observed for the specific uptake of TLQ* -IRDye800 by either colonoid. The strong fluorescence signal from the kidneys of all mice resulted from renal peptide clearance while increased intensity in the liver and lungs occurred from abundant vasculature. Low peptide uptake was observed in the other organs.

### 3.9. Peptide Safety

Vital organs were excised, sectioned, and evaluated by histology (H&E) at 48 h after intravenous injection of QQT* -IRDye800. No signs of acute toxicity were found in animal necropsy (Appendix A). Whole blood was collected for the evaluation of hematology and chemistry. No significant abnormalities were found (Appendix A).

## 4. Discussion

Here, we demonstrated the in vivo use of patient-derived colonic organoids implanted in the colon of immunocompromised mice as a preclinical model system to evaluate the specific uptake of a NIR-labeled 7mer peptide specific for cMet. Human colonoids provide clinically relevant levels of target expression. The orthotopic location generates representative vasculature to appropriately evaluate ligand delivery. The fluorescence signal peaked at 1 h following intravenous peptide injection and returned to baseline by ~24 h. This time frame was consistent with rapid renal clearance of the ligand. The peptide was found to be stable in serum with a half-life of 3.6 h, which is an adequate timespan for diagnostic imaging procedures. A high correlation was observed from the co-staining of peptide and anti-cMet antibody to cells and tissues to support ex vivo validation of binding to the intended target. A human-specific cytokeratin stain confirmed the presence of patient-derived tissues within normal mouse colonic mucosa in ex vivo analysis. The peptide biodistribution assessed ex vivo at 1 h post-injection confirmed strong uptake by the adenoma of the cMet peptide. No signs of acute toxicity were found in animal necropsy, and no significant abnormalities were found in serum hematology and chemistry. These results demonstrate potential for QQT* -IRDye800 to detect premalignant colonic lesions by generating strong fluorescence contrast as an adjunct to conventional white light illumination.

The human organoids used in this study were derived from normal and transformed patient tissues and were cultured in vitro to recapitulate the tumor development process [39,40,41]. We have previously developed efficient orthotopic transplantation methods to enable physiological tumor growth in preclinical cancer models [36] and are now using this technique to advance the clinical translation of novel imaging methodologies. Human colonoids have been established from patient-derived normal tubular adenomas and sessile serrated adenomas (SSA). Human genomic expression has been authenticated and their genetic variations have been characterized. By implanting colonoids orthotopically, repetitive in vivo imaging was performed to characterize the time frame for in vivo tumor uptake and clearance. A small animal endoscopy was adapted to collect NIR fluorescence in a spectral regime with minimal autofluorescence background. This instrument was used repetitively in the mouse colon without trauma. The same mice were used to evaluate both the target and control peptides after a washout period of 48 h to achieve statistical rigor using a minimum number of animals. The results of these studies provide in vivo evidence to support moving forward with more costly, time-consuming human clinical studies.

Moreover, human colonoids provide clinically relevant target expression levels that are representative of the patient population. Tumors were implanted in the orthotopic location to generate relevant vasculature to characterize ligand delivery. Also, this model offers a more accurate tumor microenvironment to evaluate the in vivo peptide uptake. The sequence was scrambled to provide a rigorous control to confirm specific ligand binding to cMet. The greater fluorescence intensities collected from the lesions with the target versus control peptide cannot be explained by an enhanced permeability and retention (EPR) effect alone. Also, the fluorescence signals measured were sensitive to the cMet expression level (cMet^high^ versus cMet^low^) to further support specific target binding. These organoids were used to consider heterogeneity in cMet expression seen in the clinic. In addition, this methodology was used to identify spontaneous adenomas in *CPC;Apc* mice, a genetically engineered mouse model. These premalignant lesions presented in vivo with different morphologies, including flat, small, and grossly visible. Many of these lesions were barely distinguishable with white light illumination alone. These results support future clinical translation of this methodology to identify flat and subtle premalignant lesions that can be easily missed using conventional white light colonoscopy.

Previously, a Cy5-labeled 26mer cyclic peptide that binds cMet overexpressed by colonic adenomas was intravenously administered to patients undergoing a routine colonoscopy [19]. This first-in-human pilot study demonstrated the feasibility for a fluorescence colonoscopy using a NIR-labeled peptide to detect adenomas missed by a conventional white light colonoscopy. However, the in vivo images collected demonstrated high nonspecific binding activity as reflected by a strong fluorescence background. Thereafter, the same ligand was labeled with ^18^F for PET imaging, and an ~2-fold higher uptake was found in tumor versus mammary fat pad after 10 days in a breast cancer xenograft model [42]. In another study, a 12mer cMet peptide was also labeled with ^18^F for PET imaging and demonstrated high tumor uptake in a human xenograft head and neck squamous cell carcinoma (HNSCC) model [43]. Previously, we have fluorescently labeled a 7-mer cMet peptide with Cy5.5 and demonstrated in vivo imaging in a preclinical model of spontaneous CRC using topical administration [38].

In our study, we used a smaller peptide sequence consisting of only 7 amino acids and found a high mean T/B ratio > 2.4 for patient-derived adenoma colonoids in vivo. A separate study demonstrated fluorescence-guided detection of colorectal adenomas using IRDye800-labeled bevacizumab [15]. This antibody is specific for vascular endothelial growth factor A (VEGFA), which is upregulated in colonic adenomas. This fluorescently labeled antibody was intravenously administered in patients with familial adenomatous polyposis. While the results were promising, antibodies are much more costly ligand platforms versus peptides, and the expense is difficult to justify for diagnostic use. Also, antibodies have much longer circulation times leading to a delayed peak uptake (days versus hours), and are immunogenic, which limits the ability for repeat use.

## 5. Conclusions

Patient-derived adenoma and normal colon organoids were implanted orthotopically in the colon of immunocompromised mice. Flat and subtle lesions were formed with clinically relevant expression levels of cMet. A human preclinical model system has been demonstrated to characterize in vivo uptake of a NIR-labeled peptide ligand to localize premalignant colonic lesions in vivo using endoscopic imaging by generating high fluorescence contrast.

## Figures and Tables

**Figure 1 cancers-15-04795-f001:**
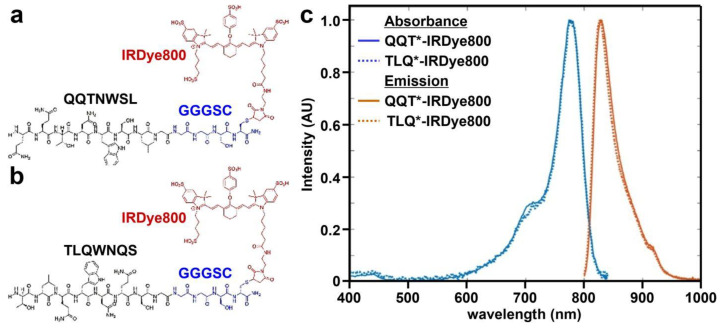
Peptide specific for cMet. (**a**) QQTNWSL and (**b**) TLQWNQS (scrambled control) are labeled with IRDye800 (red) via a GGGSC linker (blue). (**c**) Peak absorbance and fluorescence emission wavelengths are identified at λ_abs_ = 785 and λ_em_ = 825 nm, respectively.

**Figure 2 cancers-15-04795-f002:**
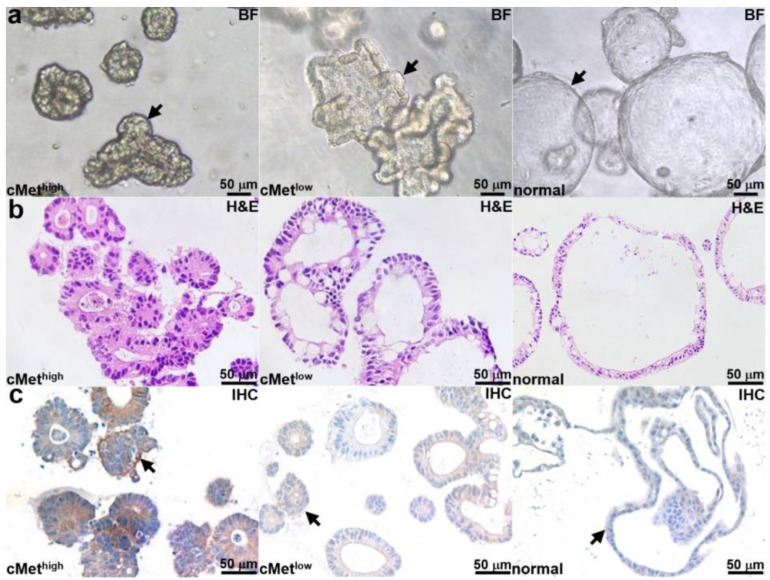
Colonoid characterization. (**a**) Brightfield (BF) images are shown for patient-derived adenoma colonoids (arrows) with high and low cMet expression and for a normal colonoid. (**b**) H&E stains of colonoid sections are shown. (**c**) IHC stain confirms high and low cMet expression (arrows) in the adenoma colonoid and low cMet expression (arrows) in the normal colonoid.

**Figure 3 cancers-15-04795-f003:**
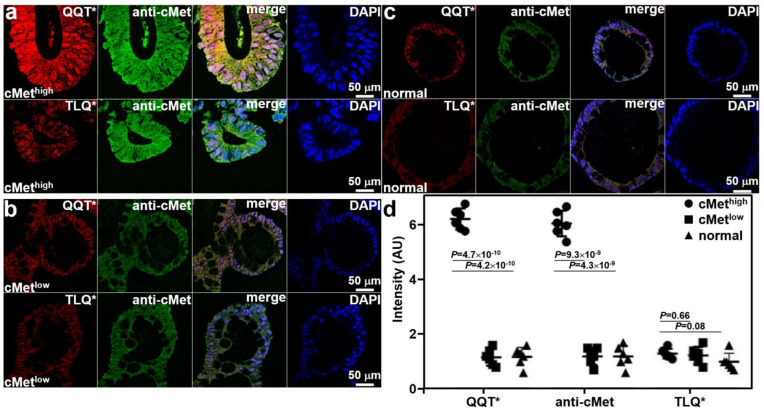
Validation of specific peptide binding ex vivo. (**a**) QQT*-IRDye800 (red) and anti-cMet-AF488 (green) showed strong binding to cMet^high^ adenoma colonoid while TLQ*-IRDye800 (control) showed minimal binding. (**b**) QQT*-IRDye800 (red), anti-cMet-AF488 (green), and TLQ*-IRDye800 showed reduced signal to cMet^low^ colonoid. (**c**) QQT*-IRDye800 (red), anti-cMet-AF488 (green), and TLQ*-IRDye800 showed minimal intensity to normal colonoid. Merged images showed co-localization of QQT*-IRDye800 and anti-cMet-AF488. DAPI stain showed the presence of cell nuclei. (**d**) Quantified fluorescence intensities from *n* = 6 colonoids in each group showed a mean (±SD) value for QQT*-IRDye800 and anti-cMet-AF488 (6.25 ± 0.38 and 6.08 ± 0.47) for the cMet^high^ adenoma colonoid. The results for the cMet^high^ adenoma colonoid were significantly greater than those for the cMet^low^ adenoma (1.17 ± 0.30 and 1.20 ± 0.31) and normal colonoids (1.18 ± 0.35 and 1.20 ± 0.39). The differences in mean intensities for TLQ*-IRDye800 (1.30 ± 0.18, 1.23 ± 0.31, and 1.00 ± 0.32) were nonsignificant. *p*-Values were calculated by a two-sample paired *t*-test. Pearson’s correlation coefficient values of r = 0.82, 0.71, 0.80, 0.79, 0.74, and 0.72, respectively, were calculated from the merged images.

**Figure 4 cancers-15-04795-f004:**
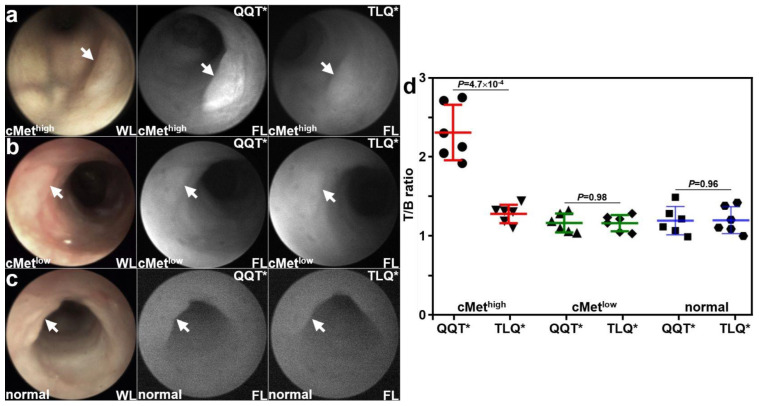
In vivo validation with patient-derived colonoids. White light (WL) images showed viable lesions (arrows) from patient-derived adenoma and normal colonoids implanted in the NSG mouse colon. (**a**) For the cMet^high^ adenoma (arrow), QQT*-IRDye800 shows strong fluorescence (FL) intensity at 1 h post-injection compared with that for TLQ*-IRDye800 (control) from the same animal collected 48 h later. (**b**) For the cMet^low^ adenoma (arrow), the fluorescence intensities from either peptide were reduced. (**c**) For the normal colonoid (arrow), minimal fluorescence intensity was observed for either peptide. (**d**) Quantified fluorescence intensities showed a significantly greater mean (±SD) value for QQT*-IRDye800 versus TLQ*-IRDye800 (2.35 ± 0.35 versus 1.31 ± 0.12) for the cMet^high^ adenoma, representing a 1.8-fold increase. There was no significant difference for the cMet^low^ adenoma (1.19 ± 0.12 versus 1.19 ± 0.11) and normal colonoid (1.22 ± 0.18 versus 1.22 ± 0.17). *p*-Values were calculated by the two-sample paired *t*-test. *n* = 6 mice were imaged in each group.

**Figure 5 cancers-15-04795-f005:**
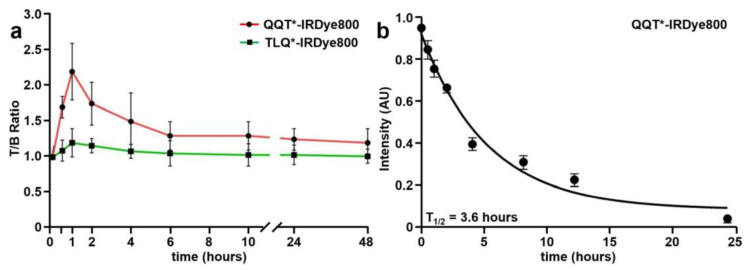
In vivo peptide properties. (**a**) T/B ratios were measured between 0–48 h from *CPC;Apc* mice following intravenous injection of QQT*-IRDye800 (200 μM, 200 μL). Peak uptake by the adenoma was observed with a mean (±SD) value of 2.2 ± 0.4 at 1 h post-injection with clearance by ~24 h. (**b**) A half-life of T_1/2_ = 3.6 h with R^2^ = 0.99 was measured in mouse serum for QQT*-IRDye800. A total of *n* = 6 mice were evaluated in each group.

**Figure 6 cancers-15-04795-f006:**
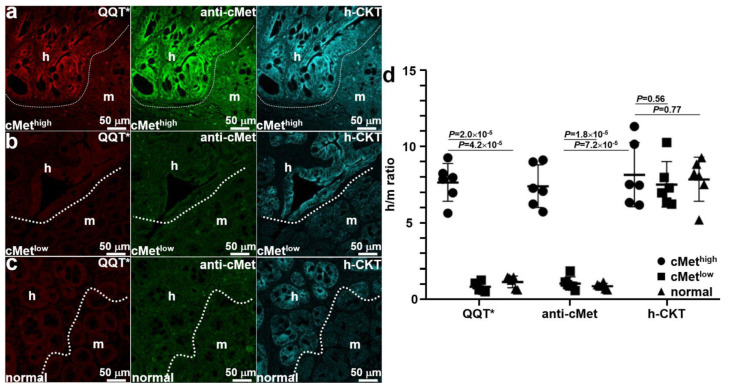
Ex vivo validation of implanted patient-derived colonoids. (**a**) The strong fluorescence signal was seen from QQT*-IRDye800 (red) and anti-cMet-AF488 (green) in the human (h) versus mouse (m) regions of an excised specimen of patient-derived cMet^high^ adenoma implanted in the NSG mouse colon. Bright intensity from antihuman cytokeratin (h-CKT, cyan) defines the boundary (dotted line) that separates the two regions. For the (**b**) cMet^low^ adenoma and (**c**) normal colonoids, reduced fluorescence intensity was seen for QQT*-IRDye800 (red) and anti-cMet-AF488 (green) from both regions. (**d**) Quantified fluorescence intensities with human-to-mouse (h/m) ratios from *n* = 6 colonoids in each group show a mean (±SD) value for QQT*-IRDye800 and anti-cMet-AF488 that was significantly greater for cMet^high^ adenoma colonoid (7.68 ± 1.23 and 7.43 ± 1.39) versus the cMet^low^ adenoma (0.87 ± 0.31 and 1.08 ± 0.43) and normal colonoid (1.19 ± 0.39 and 0.90 ± 0.19). The differences in mean intensities for antihuman cytokeratin (8.19 ± 2.10, 7.55 ± 1.48, and 7.87 ± 1.46) were nonsignificant. *p*-Values were calculated by the two sample unpaired *t*-test.

## Data Availability

The reported data, including the Appendix A, are available from the corresponding authors upon request.

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
