# Peer review of "Near-Infrared Imaging of Colonic Adenomas In Vivo Using Orthotopic Human Organoids for Early Cancer Detection"

_cancers, 2023, doi:10.3390/cancers15194795_

Round 1

Reviewer 1 Report

The authors underscore the significance of utilizing patient-derived colonoids as a valuable pre-clinical model to evaluate the effectiveness of a near-infrared peptide in detecting pre-malignant colonic lesions in vivo. Their research findings showcased a peak target-to-background ratio approximately one hour after intravenous peptide administration, followed by complete signal clearance within 24 hours. Notably, the peptide demonstrated stability in serum, characterized by a half-life of 3.6 hours. The authors conducted co-staining experiments on adenoma and normal colonoids, revealing a robust correlation between the peptide and anti-cMet antibody in ex vivo settings. Additionally, human-specific cytokeratin staining was employed to confirm the presence of human tissue within the normal mouse colonic mucosa. The peptide's biodistribution indicated rapid renal clearance, and no signs of acute toxicity were observed.

This research holds promise for improving early diagnosis and intervention in colorectal cancer. The study was well-executed and is a strong candidate for publication, although minor comments for improvement have been noted.

1.     Have there been any alterations in the quantum yield or fluorescence lifetime of the peptide-conjugates when compared to the parent IR dye?

2.     Is the peptide conjugate completely soluble in water? It would be beneficial to provide details regarding the exact drug formulation used for both in vitro and in vivo studies.

3.     Given the existence of several peptide-based IR or near-IR dyes, PET, SPECT, and MRI approaches in colorectal cancer imaging by other research groups, the authors should acknowledge and discuss these references in the appropriate sections, highlighting advantages and disadvantages as relevant.

Reviewer 2 Report

1-      All abbreviations need to full-term explanation (for example cMet,  ….).

2-      Did you perform any experiment under in vitro conditions for using a valid concentration of injection of the used drug?

3-      In the discussion section, more relevant works are needed to comparison of the results.

4-       There are no whole-body mice images under in vivo circumstances using Near Infrared.

5-      Minor editing of English language required

Minor editing of English language required
